# The frequency of pathogenic variation in the *All of Us* cohort reveals ancestry-driven disparities

Eric Venner [1✉], Karynne Patterson [2], Divya Kalra[1], Marsha M. Wheeler[2], Yi-Ju Chen[1], Sara E. Kalla [1], Bo Yuan[1], Jason H. Karnes [3,4], Kimberly Walker [1], Joshua D. Smith[2], Sean McGee[2], Aparna Radhakrishnan[2], Andrew Haddad[5], Philip E. Empey [6], Qiaoyan Wang[1], Lee Lichtenstein[7], Diana Toledo[7], Gail Jarvik[8,9], Anjene Musick [10] & Richard A. Gibbs[1] on behalf of the All of Us Research Program Investigators*

Disparities in data underlying clinical genomic interpretation is an acknowledged problem, but there is a paucity of data demonstrating it. The *All of Us* Research Program is collecting data including whole-genome sequences, health records, and surveys for at least a million participants with diverse ancestry and access to healthcare, representing one of the largest biomedical research repositories of its kind. Here, we examine pathogenic and likely pathogenic variants that were identified in the *All of Us* cohort. The European ancestry subgroup showed the highest overall rate of pathogenic variation, with 2.26% of participants having a pathogenic variant. Other ancestry groups had lower rates of pathogenic variation, including 1.62% for the African ancestry group and 1.32% in the Latino/Admixed American ancestry group. Pathogenic variants were most frequently observed in genes related to Breast/Ovarian Cancer or Hypercholesterolemia. Variant frequencies in many genes were consistent with the data from the public gnomAD database, with some notable exceptions resolved using gnomAD subsets. Differences in pathogenic variant frequency observed between ancestral groups generally indicate biases of ascertainment of knowledge about those variants, but some deviations may be indicative of differences in disease prevalence. This work will allow targeted precision medicine efforts at revealed disparities.

[1] Human Genome Sequencing Center, Baylor College of Medicine, Houston, TX, USA. [2] Department of Genome Sciences, University of Washington, Seattle, WA, USA. [3] University of Arizona, R Ken Coit College of Pharmacy, Department of Pharmacy Practice and Science, Tucson, AZ, USA. [4] Vanderbilt University Medical Center, Department of Biomedical Informatics, Boston, MA, USA. [5] Department of Pharmaceutical Sciences, University of Pittsburgh School of Pharmacy, Pittsburgh, PA, USA. [6] Department of Pharmacy and Therapeutics, University of Pittsburgh School of Pharmacy, Pittsburgh, PA, USA. [7] Broad Institute of MIT and Harvard, Cambridge, MA, USA. [8] Department of Medicine (Medical Genetics), University of Washington School of Medicine, Seattle, WA, USA. [9] Department of Genome Sciences, University of Washington School of Medicine, Seattle, WA, USA. [10] NIH All of Us Research Program, National Institutes of Health Office of the Director, Bethesda, MD, USA. *A list of authors and their affiliations appears at the end of the paper. ✉email: venner@bcm.edu

mplementing genomic medicine will require interpreting genomic variation in real-world clinical populations. At present, lack of diversity in large genomics cohorts is a widely recognized problem[1–4]. Most sequencing studies have focused on European ancestry populations[5] and it is predicted that much of the pathogenic variation present in the general population is specific to an ancestral population[6,7]. Overcoming this gap in diagnostic yield will necessitate collecting diverse genomic data paired with electronic health record data[8,9].

To advance precision medicine, the *All of Us* Research Program from the National Institutes of Health (NIH) is generating a unique dataset which includes genetic, electronic health record and survey data from a diverse participant cohort[10]. *All of Us* is targeting a cohort size of more than one million, with a focus on individuals who have been traditionally underserved by biomedical research[10]. Whole genome sequence data is generated at one of three *All of Us* Genome Centers located at the Baylor College of Medicine, the Broad Institute and the Northwest Genomics Center at the University of Washington. Data are transferred via the *All of Us* Data and Research Center at Vanderbilt University to Clinical Validation Laboratories at the Baylor College of Medicine, Northwest Genomics Center and Color Genomics, where sequence data are interpreted for health-related reporting. Data are also deposited at the Data Resource Center for further processing and sharing with qualified researchers via the *All of Us* Researcher Workbench, a cloud-computing platform. The data generation and results workflow are conducted under an investigational device exemption through the FDA[11].

A strength of the *All of Us* Research Program is the diversity of the participants, relative to those in previously studied large cohorts. The *All of Us* Researcher Workbench, a cloud computing platform, contains whole-genome sequencing data from 98,590 participants (https://doi.org/10.1038/s41586-023-06957-x) in a data release called alpha3. Based on genetically predicted ancestry (see Methods), this dataset included 49,668 (50.4%) participants with predominantly European ancestry, 22,897 (23.2%) participants with predominantly African ancestry, 15,893 (16.1%) Latino/ Admixed American ancestry participants, 2,113 (2.1%) East Asian ancestry participants, 940 (1.0%) South Asian ancestry participants, 193 (0.2%) Middle Eastern participants and 6,886 (7%) participants which do not group unambiguously and were designated as Other. In contrast, the UK Biobank project contains 94% European ancestry individuals, the Million Veterans project contains 77% and eMERGE contains 73%.[5]

However, it is currently unknown whether the frequency of pathogenic variants in genes conferring appreciable health risks in the *All of Us* cohort will differ from those in previously ascertained healthy populations due to the unprecedented diversity of participants enrolled in the program. Identification of such differences will be a powerful reinforcement of the importance of the program's strategy for recruitment and engagement of participants from underrepresented groups.

To examine the frequency of pathogenic genomic variation in participants we examined data from a set of 73 genes that harbor actionable secondary findings[12]. These genes are associated with diseases, including hereditary breast cancer, hemochromatosis, dislipidemias and cardiomyopathies and represent some of the most well-studied targets for genomic medicine[12]. We annotated each participant with their calculated genetic ancestry and searched for known pathogenic variants with established criteria for pathogenicity, aided by databases of curated variants from previous clinical genomic projects[13,14], as well as an early de-identified review of *All of Us* data. The preliminary results from analysis of data from more than 98,000 *All of Us* participants showed variability in the rates of pathogenic variants between ancestry groups, prompting further analysis of the source of the differences.

## Results

**Rates of pathogenic variation in the all of Us dataset.** To understand the rates of previously-known pathogenic variants broken down by predicted ancestry groups in the *All of Us data*, we used the 'VIP' database to annotate P/LP variants[15] present in whole genome sequencing data in *All of Us* participants across 73 genes with actionable secondary findings[12]. This database contains variants curated by the HGSC-CL variant interpretation group during projects such as eMERGE III[13] and HeartCare[14] as well as an initial assessment of de-identified variants from the *All of Us* cohort itself. Figure 1a shows that the European ancestry group has the highest rate of previously known pathogenic variants (2.13%), followed by the "Other" group at 1.82% and the African ancestry group at 1.52%. Using a Chi-square test for independence, the rates of pathogenic variants are significantly different between the African ancestry, European and Admixed American/Latino groups ($p < 0.00001$, $n = 1757$ total observed P/LP variants, Supplementary Table 1), which have sufficient data to perform this test. These differences are significant even when leaving out the gene *HFE*, which has a large known difference in pathogenic rates between ancestry groups. These significant differences in the rates of pathogenic variation could be explained either by a bias in the ascertainment of pathogenic variation in the variant database or by differences in the underlying disease prevalence between ancestry groups.

To gain a more comprehensive understanding of pathogenic and likely pathogenic variants in participants, we incorporated rare (GnomAD popmax allele frequency below 0.001), predicted loss-of-function (pLoF) variants into the prior analysis of known pathogenic variants (Supplementary Table 2). We focused on 38 specific genes where loss-of-function is a recognized cause of disease, and we also looked for any overlap with known pathogenic variation (refer to Fig. 1a, orange and gray bars). Our findings revealed 1114 variants, comprising 562 frameshifts, 112 splice acceptor variants, 100 splice donor variants, and 340 stop gain variants (see Supplementary Table 3 for details). Many of these pLoF variants overlap the pathogenic/likely pathogenic variants in the VIP database (Fig. 1a, b, orange bars). For example, in the group with European ancestry, the overlap was 0.46%. Including pLoF variants in our analysis increases the overall rates of positive variants, ranging from 2.26% in the European ancestry group to 1.32% in the Latino / Admixed American group, with smaller ancestry groups having wider confidence intervals.

When looking purely at rates of rare pLoF variants, the European ancestry group had a lower rate of findings than both the South Asian ancestry group (0.85%) and the East Asian ancestry group (0.62%), though this may be due to using GnomAD, which contains samples from predominantly European ancestry, as a filter. The variation of pLoF variants between different ancestry groups was less than that of the pathogenic/likely pathogenic variants (pLoF standard deviation of variant rates was 0.003, versus a standard deviation of 0.004 for P/LP variant rates). Given that interpreting novel LoF variants can be simplified and automated, these findings suggest that the detection of pathogenic variants in studies with a large number of participants with European ancestry may be contributing to some of the differences observed between groups.

To evaluate whether known variants with ancestral divergence are replicated in the All of Us cohort, we examined the frequency of the rs334 mutation in *HBB*, known to be associated with sickle cell disease, and the *APOL1* G1 and G2 alleles (Table 1). The results confirm the expected ancestral divergences, with non-reference alleles appearing 17,969 times within the 22,897 participants of African ancestry, and 68 times among the 49,668 participants of European ancestry.

In order to understand which genes have divergent rates of pathogenic variants between ancestry groups, we normalized

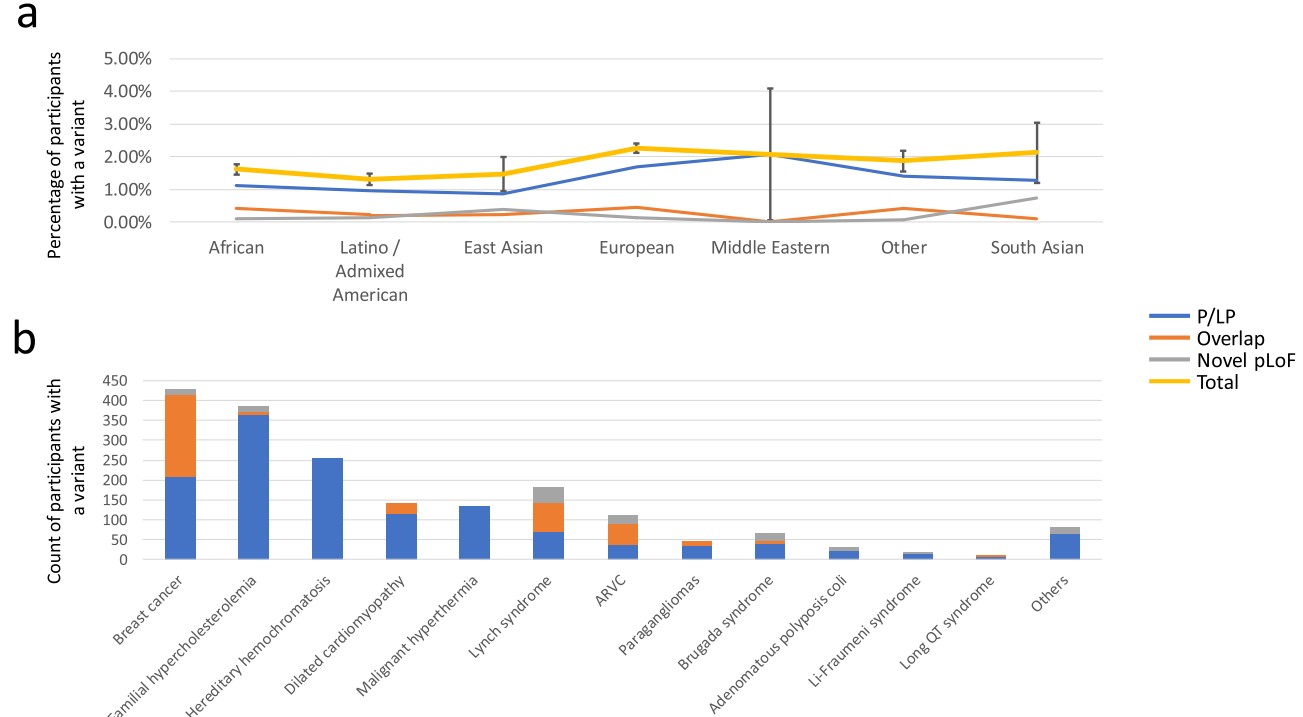

**Fig. 1 Pathogenic variants by ancestry.** Using a database of known pathogenic mutations and annotations for rare, pLoF variants, we searched the beta release of the All of Us cohort for pathogenic variants, on the Researcher Workbench. Figure 1**a** shows the rates of pathogenic variation, broken down by predicted genetic ancestry groups. Error bars show 95% the confidence intervals for the total set of pathogenic variants (including both VIP P/LP variants and rare pLoF). Figure 1**b** shows the breakdown of pathogenic variants by disease area. The blue line and bar depict the rate of Pathogenic and Likely pathogenic variants, the gray bar the rate of novel, predicted loss of function variants and the orange bar depicts predicted loss of function variants that were known pathogenic variants at the time of analysis. The yellow line shows the total variants in each ancestry group.

---

**Table 1 Non-reference sample counts in known ancestrally divergent genes.**

|  | African | Latino / Admixed American | East Asian | European | Middle Eastern | Other | South Asian |
|---|---|---|---|---|---|---|---|
| *APOL1* (G1 & G2 alleles) | 17,969/22,897 (78.48%) | 1041/15,893 (6.55%) | 1/2113 (0.05%) | 60/49,668 (0.12%) | 1/193 (0.52%) | 1166/6886 (16.93%) | 2/940 (0.21%) |
| *HBB* (rs334) | 2059/22,897 (8.99%) | 229/15,893 (1.44%) | 0/2113 (0.00%) | 6/49,668 (0.01%) | 0/193 (0.00%) | 202/6886 (2.93%) | 2/940 (0.21%) |

To confirm the presence of known alleles with ancestral divergence, we examined the *APOL1* G1 and G2 alleles, as well as the rs334 SNP in the *HBB* gene. The counts in the table above include all instances of the rs73885319, rs60910145 SNPS (*APOL1* G1 allele) as well as the G2 deletion (rs71785313) and the rs334 allele in the *HBB* gene, associated with Sickle-cell disease. Participants may carry more than one allele. These variants are present at a higher rate in participants with African ancestry, as expected.

---

pathogenic rates of all genes against the European ancestry group's pathogenic rate and checked for outliers in population proportions using a Bonferoni-corrected z-test. Several genes show significant differences from the European ancestry group's rates (Table 2). Interestingly, the genes *HFE* and *PALB2* differences are known[16,17] but the differences in the gene *PKP2* have not been reported previously to our knowledge. *APOB* shows differences that may indicate altered sources of genetic disease prevalence in some ancestry groups. In the African ancestry subgroup, the *PALB2* and *PKP2* findings were replicated using ClinVar as a source of pathogenic variants instead of the VIP database (Supplementary Table 4). These gene-level differences present targets for future investigations of health disparities.

**Comparison with GnomAD.** In order to understand how these findings compare to other large cohorts, we compared these rates of pathogenic findings to gnomAD. To do so, we first identified pathogenic variation using the VIP database and ClinVar

separately. We then annotated these variants with the allele frequencies from the *All of Us* cohort and gnomAD, and summed up the variants in genes to provide gene-level summaries. Under the hypothesis that gnomAD may contain affected individuals, we also made selected comparisons to either the gnomAD non-cancer subset for cancer genes or the non-TopMed subset for genes related to cardiovascular disease. The results (Fig. 2) show the relative difference between *All of Us* and gnomAD positive rate, broken down by gene. Overall, we observe high-level concordance between these data sources (Pearson correlation 0.99 between *All of Us* and gnomAD positive rates). However, when comparing gene-level pathogenic rates in ancestry groups other than the European ancestry group, some differences are apparent. For example, in the African ancestry group, the incidence of P/LP *BRCA2* variants differs between *All of Us* cohort and gnomAD (0.093% vs 0.161%, Fig. 3a). However, gnomAD is available in different subsets[18], and when the non-cancer subset is used, the pathogenic rate is very similar to that in the *All of Us*

**Table 2 Genes having rates of pathogenicity that differ from the European pathogenic rate.**

| Gene | Ancestry group | Ancestry group path. variants | European group path. variants | *p* value |
|------|----------------|-------------------------------|-------------------------------|-----------|
| *APOB* | African | 4/22,897 (0.02%) | 57/49,668 (0.11%) | 0.0001 |
| *PKP2* | African | 33/22,897 (0.14%) | 20/49,668 (0.04%) | 0.00001 |
| *PALB2* | African | 33/22,897 (0.14%) | 30/49,668 (0.06%) | 0.0014 |

Pathogenic rates in each gene were compared to the European rate, with significant deviations in population proportions detected with a Bonferroni-adjusted z-test. *All of Us* allele frequencies are based on biologically independent samples.

cohort (0.093% vs 0.095%). A similar story is found with the *LDLR* gene in the Admixed American/Latino ancestry group: using a broad comparison between gnomAD and *All of Us*, the rates of pathogenicity are divergent (Fig. 3b). However, when using the gnomAD non-TopMed subset, the rates of pathogenicity are highly similar, possibly due to dyslipidemia studies in TopMed[19]. Other examples are not as clear. For example, we note divergent pathogenicity rates in the *RYR1* gene between the gnomAD and *All of Us* African ancestry groups, but could not account for them using a gnomAD subset. Overall, these results show that the rate of pathogenic variant findings in the *All of Us* are similar to those seen in gnomAD, and the rates become even closer when we account for disease populations within the gnomAD resource.

**Comparing with eMERGE III**. In further analysis, we examined the rate of pathogenic variants as seen in the eMERGE III project. This program involved eleven clinical sites providing samples to two clinical laboratories for analysis and reporting. The network utilized a custom gene panel of 109 genes, incorporating the American College of Medical Genetics 59 secondary findings list along with clinical site-specific genes. Notably for this comparison, a substantial fraction of the samples from this program were not chosen based on specific diseases. The observations in the All of Us cohort align with expectations drawn from pathogenic variant frequency in the Association of Molecular Pathologists/American College of Medical Genetics secondary findings genes (See Supplementary Table 5, correlation coefficient for the rate of findings is 0.78). Overall, the eMERGE III cohort exhibited a higher incidence of pathogenic findings. This can possibly be attributed to the different genes present in the gene panel and the fact that the set of genes that were returned varied among clinical sites. Moreover, the eMERGE data underwent complete review, unlike the All of Us data, where we are citing known pathogenic variants. There is a remarkably close match for hemochromatosis, which had the same reporting criteria for the homozygous pathogenic allele C282Y. The substantial discrepancy in clotting disorders is likely due to the decision to report the F5 Leiden variant in eMERGE III.

**Assessing the potential for participant selection effects**. As an additional approach to understand whether *All of Us* participants with known genetic diseases are self-selecting for participation in the program, which could impact the counts of pathogenic variants impacting Mendelian disease. We selected four rare and four more common variants associated with disease and examined their allele frequencies relative to that in GnomAD (Table 3). To best match the *All of Us* ancestry groups, the GnomAD Eur group was taken as a combination of Finnish, non-Finnish and Ashkenazi populations. For these variants, we observe a close match between their GnomAD and *All of Us* allele frequencies. Only the common *HFE* rs1800562 alleles, in which a 6% difference was found, was significantly different (Bonferoni corrected critical value of 0.00625, $n = 82,106$ biologically independent participant samples). Based on

this analysis, we do not find an appreciable amount of ascertainment bias in this population.

**Enriching for specific genetic factors**. When broken down by disease area, the results show that the predominant health-related findings will be in breast cancer, familial hypercholesterolemia, dilated cardiomyopathy and hereditary hemochromatosis (Fig. 1b). To further understand how these findings relate to the participant's available health information, we made use of additional data resources provided in the AoU Researcher Workbench. The *All of Us* Research Workbench provides participant health information in the forms of electronic health record condition codes as well as survey questions answered by the participants. To begin to understand how this phenotypic information available in the workbench matches with genetic findings, we looked at breast cancer as an example. Samples were selected if participants had either "Malignant neoplasm of female breast" or "Malignant tumor of breast" condition codes (SNOMED Codes 254837009 and 372064008) or if the participant answered "breast cancer" to the question "Has a doctor or health care provider ever told you that you have or had any following cancers?" 8603 participants fell into this cohort, with 1653 of those having whole-genome sequencing data thus far. This represents an enrichment for P/LP variants in breast cancer patients (32/1653, 1.94% vs 414/98,590, 0.42%, $p < 0.00001$), demonstrating the ability to use *All of Us* participant level data to select cohorts enriched for specific genetic factors.

**Discussion**

The diverse cohort collected as part of the *All of Us* will be a rich resource for advancing precision medicine. Here, we have examined the pathogenic variant rate in the beta release of the *All of Us* Researcher Workbench data, finding significant variability between groups of participants with differing ancestry. This variability is likely the result of multiple factors, but ascertainment of pathogenic variants in databases is likely to contribute substantially. Future work will show whether variant interpretation of the *All of Us* diverse cohort will have an impact on this ascertainment bias of pathogenic variants, but future targeted efforts that aim to perform clinical interpretation of non-European participants could also be necessary.

Ancestry-linked differences in variant pathogenicity found in different genes highlight the importance of the *All of Us's* diverse cohort. Different ancestry backgrounds likely carry different burdens of risk depending on the frequency of pathogenic haplotypes. Implementing precision medicine will require understanding these varied risk profiles and gathering detailed information on the haplotype structure of the population. Clinical testing is guided to some extent today by ancestry, for example, *BRCA2* in the Ashkenazi Jewish population[20] and HLA-B testing for SJS/TEN in some Asian populations[9,21,22]. As we better understand genetic risk burdens in population groupings we can more precisely target genetic testing and aid interpretation.

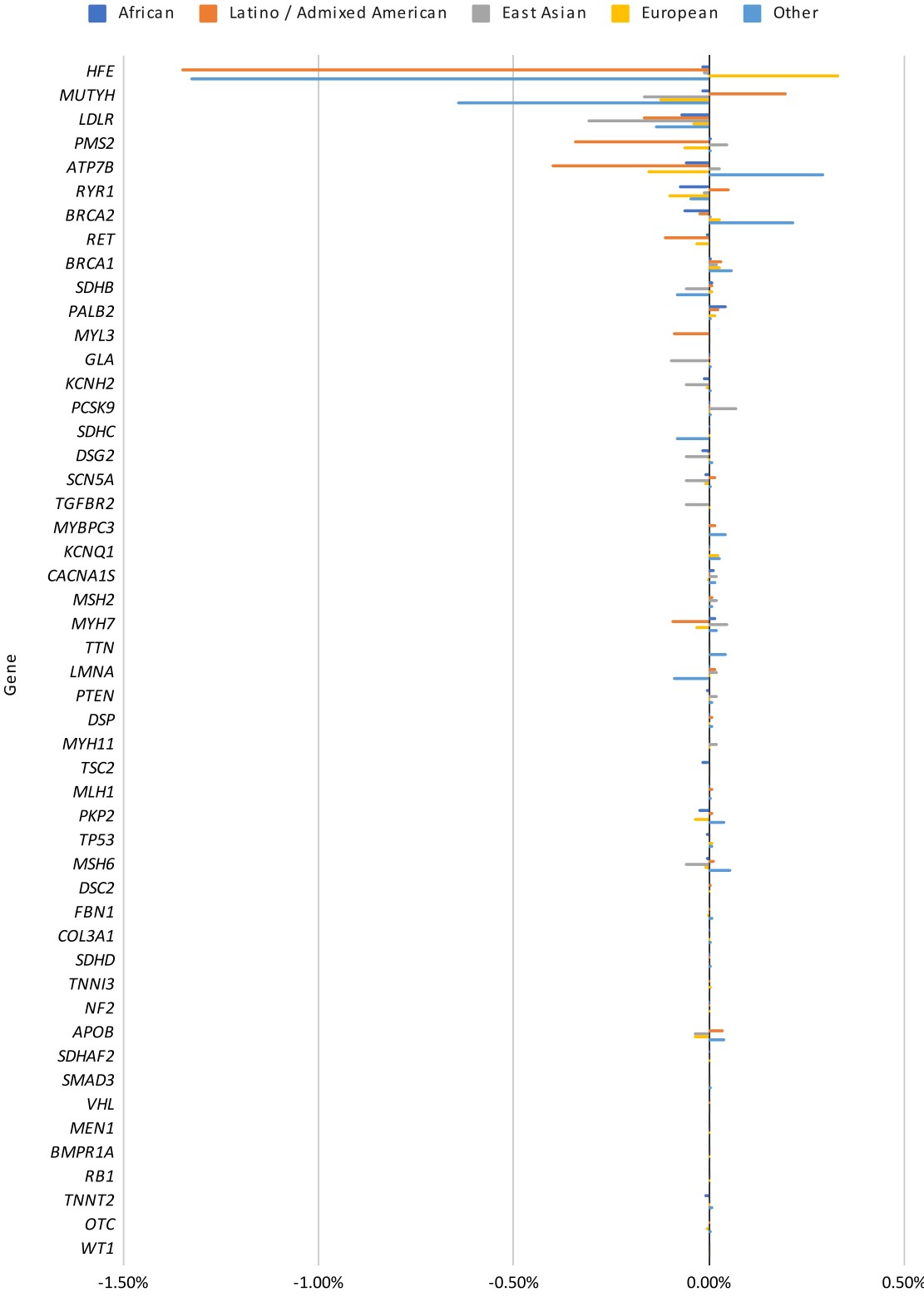

**Fig. 2 Relative positive rates for *All of Us* vs gnomAD.** This figure shows relative frequencies of previously-curated pathogenic or likely pathogenic variants between the *All of Us* cohort and gnomAD, broken down by gene and ancestry group. Overall, there is a high level of concordance between variant frequencies of pathogenic variants; most genes show very small differences relative to gnomAD. Ancestries are shown as Dark blue for African, Orange for Latino / Admixed American, Gray for East Asian, Yellow for European and light blue for Other.

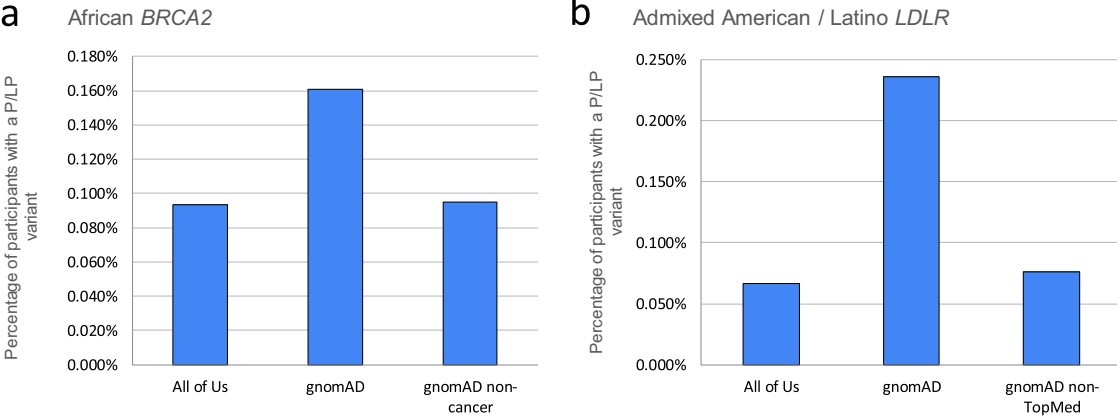

**Fig. 3 Comparisons to gnomAD subsets.** Though there is high level concordance between the rates of pathogenic variants in All of Us and gnomAD, in some cases there are differences specific to a gene and ancestry group. For example, in participants with African ancestry, the rate of pathogenic variants in *BRCA2* diverges from gnomAD (**a**). However, when the non-cancer subgroup of gnomAD is used, the rates are much more similar. A similar situation is seen in the Admixed American / Latino ancestry group with *LDLR* (**b**). Using the non-TopMed portion of gnomAD brings the rates much closer.

Another benefit of this data has been to allow the program to project forward to the "return of health-related results" phase of the program. Based on the variants examined in this work, we estimate that 17% of variant interpretations will require manual assessment of literature, which has strong implications for the time required to complete the review of that variant. This in turn has allowed us to better project the resources required to return health-related results.

We primarily used an internal database of pathogenic genomic variants in this work (the 'VIP' database) with ClinVar[23] used as a confirmatory resource. Although ClinVar is an essential and widely-used resource, its heterogeneity poses a challenge. For example, the well-known *HFE* NM_000410.4:c.845 G > A variant, which we previously reported clinically for the eMERGE III[13] project when in the homozygous state, is associated with hemochromatosis. However, at the time of publication, this variant is listed as "conflicting" in ClinVar and so was excluded under our simple ClinVar filtering scheme (see Methods). Many other variants likely fall into a similar scenario. It would be possible to fine-tune filters for ClinVar variants and reanalyze this data, which may provide a more complete picture of known pathogenic variation, but there are large effects on sensitivity and specificity that arise due to this tuning that would need to be understood. This level of curation was beyond the scope of the current study.

Ancestry estimation approaches that assign a single continental ancestry to an individual's entire genome face many issues, including failing to accurately portray the make-up of admixed individuals and potentially grouping individuals that may not share recent ancestry. In future work we expect to adopt more nuanced approaches to ancestry prediction. One such approach may be to use 'local ancestry', in which ancestry information is tracked at the variant level, with each variant assigned proportionally to one of many known ancestry groups.

It is paramount that the field address ascertainment biases in knowledge of pathogenic variants. Case-control data are a powerful tool towards achieving this, and the field would benefit greatly from creating more diverse cohorts of patients with accompanying variant interpretations and deep phenotyping, and from sharing that data widely. As new variants of unknown significance are identified, functional studies are also beneficial in their interpretation. The advent of high-throughput functional screening techniques is accelerating[24] this data collection and could be targeted at variants from underrepresented populations.

An alternative, although unlikely, explanation for the elevated rate of pathogenic variants seen in this study in the European ancestry population, is that the selected genes carry a burden of pathogenic variants that is in fact specific to individuals of European ancestry. The American College of Medical Genetics took an evidence-based approach[12] to selecting genes, prioritizing genes which, at the time, had sufficient evidence showing that patient morbidity and mortality could be reduced through genetic testing while limiting the burden on patients and clinical laboratories. Given known disparities in genetics knowledge, this process may have preferentially identified genes whose impacts on individuals with European ancestry is well studied. Remedying this will require future case-control studies that include diverse populations, which expert panels can then review as they decide on future secondary finding lists.

As a baseline for our expectations of the frequency of pathogenic variants we have compared to gnomAD[25]. At a high-level, we found strong concordance between these cohorts, although there are some outliers when individual gene-level frequencies are examined. Some differences are likely due to the cohorts applying slightly different ancestry definitions. For example, the precise definition of the "Other" group is likely to be different, and the gnomAD counts do not include the Middle Eastern and South Asian ancestry groups. Using local ancestry (i.e., assigning ancestry not at the sample-level but at the haplotype level) would help resolve this issue.

The current study faces a number of limitations. First, our variant knowledge relies on variant interpretations that were primarily carried out for other genomic reporting projects and during the lead-up to the return of health-related results for the program. As we complete the health-related return of results and the accompanying variant reviews, we will also curate new variants, which is likely to benefit the underserved populations. We are also limited by the cohort creation process adhered to by the All of Us research program. This process recruits participants through partnerships with universities, research centers, and community health centers, direct volunteers, and through community engagement. This process will undoubtedly shape the cohort; for example, we know that participants with a family history of disease are more likely to sign up for genetics studies[26]. At this time, it is not clear what effect this selection process would have on the rates of rare, pathogenic variants. As evidenced by our examination of selected known pathogenic variants and of the high-level comparisons with GnomAD, there does not appear to be a large selection bias. Further study is needed in order to understand whether there are localized or more subtle selection effects at work in this cohort.

**Table 3 Comparison of allele frequencies for specific variants with GnomAD.**

| Variant | GnomAD AF | All of Us AF | p value (Two-tailed z test) | Population |
|---|---|---|---|---|
| BRCA2 c.5946delT (p.Ser1982Argfs*22) | 0.0364% (26/71,468) | 0.0342% (34/99,336) | 0.815 | Eur |
| BRCA2 c.2808_2811delAAAG (p.Lys938Ilefs*7) | 0.0014% (2/145,080) | 0.0020% (2/99336) | 0.703 | Eur |
| BRCA2 c.8537_8538delAG (p.Ser2846Glufs*2) | 0.0021% (3/145,182) | 0.0010% (1/99,336) | 0.525 | Eur |
| LDLR c.682 G > T (p.Glu228X) | 0.0024% (2/82,108) | 0.0050% (5/99,336) | 0.375 | Eur |
| APOL1 G1 p.S342G - rs73885319 (GRCh38:chr22:36265860:A > G) | 22.27% (9208/41,338) | 22.36% (10,239/45,794) | 0.753 | Afr |
| APOL1 G1 p.I384M - rs60910145 GRCh38:chr22:36265988:T > C/G | 22.43% (9047/40,338) | 21.94% (10,045/45,794) | 0.081 | Afr |
| HBB rs334 | 4.34% (1799/41,432) | 4.50% (2,59/45,794) | 0.269 | Afr |
| HFE rs1800562 | 6.05% (4964/82,106) | 6.41% (6371/99,336) | 0.001 | Eur |

To evaluate whether self-selection by participants with known genetic diseases impacts our study, we examined the allele frequencies of eight disease-associated variants (four rare, four common) in comparison with GnomAD. GnomAD and All of Us allele frequencies closely match those in GnomAD. Except for a 6% difference in the common HFE rs1800562 alleles, the frequencies in our study closely match those in GnomAD. GnomAD and All of Us allele frequencies are based on biologically independent samples.

A further limitation is in the high-throughput nature of the pLoF analysis. Mirroring the Association of Molecular Pathologists/American College of Medical Genetics PVS1 guidelines, we filtered pLoF variants by their allele frequency, keeping only those that are rare in the GnomAD database. However, this potentially introduces a bias, as GnomAD is overrepresented with European ancestry individuals. There may remain pLoF variants that would be removed from other ancestry groups with more data. Though it is currently a limitation, as diverse datasets become available, these high-throughput estimates will improve. Although we have not been able to detect selection bias in the cohort in this analysis, the cohort likely does contain both enrichments and depletions of pathogenic variants in specific disease genes due to self-selection of the participants. These effects may be local to specific disease areas or specific subpopulations within the cohort. Researchers making use of this resource should be aware of this potential bias. Finally, although the full cohort for the All of Us project is anticipated to be the most diverse genetic resource available, the current release still features a large proportion of participants of European ancestry. Future releases will enable further exploration beyond what is possible in the current release.

The *All of Us* Researcher Workbench enabled this first assessment of pathogenic variants within the *All of Us* cohort, and the diversity of that population has allowed us to begin to detect different frequencies in pathogenic variation between several of the ancestry groups. More work in this area will further reveal groups of participants who carry under-studied pathogenic variation and allow us to target precision medicine efforts at those disparities.

## Methods
**All of Us demonstration Projects**. The *All of Us* research progra recruits participants that have been underrepresented in biomedical research through a network of affiliated HPOs and direct volunteers[27]. Demonstration projects were designed to describe the cohort, replicate previous findings for validation, and avoid novel discovery in line with the program values to ensure equal access by researchers to the data[28].The work described here was proposed by Consortium members, reviewed and overseen by the program's Science Committee, and confirmed as meeting criteria for non-human subjects research by the *All of Us* Institutional Review Board. The initial release of data and tools used in this work was published recently[28].

**All of Us research Hub**. This work was performed on data collected by the *All of Us* Research Program using the *All of Us* Researcher Workbench, a cloud-based platform where approved researchers can access and analyze *All of Us* data. The *All of Us* data currently includes surveys, electronic health records, and physical measurements. The details of the surveys are available in the Survey Explorer found in the Research Hub, a website designed to support researchers[3]. Each survey includes branching logic and all questions are optional and may be skipped by the participant. PM recorded at enrollment include systolic and diastolic blood pressure, height, weight, heart rate, waist and hip measurement, wheelchair use, and current pregnancy status. EHR data was linked for those consented participants. All three data-types are mapped to the Observational Medicines Outcomes Partnership (OMOP) common data model v 5.2 maintained by the Observational Health Data Sciences and Informatics collaborative. To protect participant privacy, a series of data transformations were applied. These included data suppression of codes with a high risk of identification such as military status; generalization of categories, including age, sex at birth, gender identity, sexual orientation, and race; and date shifting by a

random (less than one year) number of days, implemented consistently across each participant record. Documentation on privacy implementation and creation of the Curated Data Repository is available in the *All of Us* Registered Tier Data Dictionary[4]. The Researcher Workbench currently offers tools with a user interface built for selecting groups of participants (Cohort Builder), creating datasets for analysis (Dataset Builder), and Workspaces with Jupyter Notebooks to analyze data. The notebooks enable use of saved datasets and direct query using R and Python 3 programming languages.

**Annotation of known pathogenic variants**. We used the 'VIP' database to annotate known pathogenic or likely pathogenic (P/LP) variants. The VIP database is a collection of variants compiled during clinical reporting activities carried out at the Human Genome Sequencing Center-Clinical Laboratory (Supplementary Data 1). These variants were manually assessed by a team of variant curation experts led by board-certified clinical geneticists for previous Human Genome Sequencing Center-Clinical Laboratory projects, such as the NIH's eMERGE III program[13] and HeartCare[14], a local cardiovascular risk assessment project. All the variants were interpreted based on the guidelines provided by the Association for Molecular Pathology/American College of Medical Genetics and Genomics[29], as well as the most recent ClinGen recommendations. The VIP database includes 59,405 genomic variants. These variants are classified into different categories such as pathogenic, likely pathogenic, variants of uncertain significance, benign, likely benign, and risk alleles. They are distributed across 8,653 genes (See the supplemental file vip_gene_count.xslx for details). One of the key advantages of the VIP database is the consistency in variant curation, as all the variants have been curated by the Clinical Variant Interpretation team at HGSC-CL using a uniform set of criteria. This contrasts with resources like ClinVar, which may include contradictory assessments of variant pathogenicity.

**Samples/dataset**. Aggregate data for this study was generated from *All of Us* participant data ($N = 98,590$) using the *All of Us* Researcher Workbench cloud computing platform. We accessed variant data (single nucleotide variants and indels) from whole-genome sequencing in the alpha3 data release provided by the *All of Us* Data Resource Center. Details regarding *All of Us* Data Resource Center's genomic pipelines can be found here (https://doi.org/10.1038/s41586-023-06957-x). *All of Us* variant data were generated using the GRCh38 human reference build and made available on the pre-production version of the *All of Us* Researcher Workbench using the Hail framework[30]. All relevant ethical regulations were followed. All participants in this study provided written informed consent. This work was approved by the Institutional Review Board (IRB) of the All of Us Research Program.

For our analyses, we subsetted the whole genome variants Hail matrix table to coding regions for the 73 genes listed in the American College of Medical Genetics Supplementary Findings v3.0 list[12]. In these 73 genes, we also included regions 2000 bp upstream and 1000 bp downstream of the coding regions to ensure we do not exclude any pathogenic variants outside of the coding regions. Variants were filtered on genotype quality (GQ > 20) and variant pathogenicity (using 'Path' or 'LPath' values in the Vip_variant_interpretation field from the VIP database). These variants were then grouped by predicted genetic ancestry and genes to build a contingency table, with heterozygous variant counts across most genes. For the autosomal recessive genes *MUTYH*, *ATP7B* and *KCNQ1*, only homozygous variants were counted. In *HFE*, only the *NM_000410.4:c.845 G > A* variant in the homozygous

state was considered. We annotated *All of Us* participants with predicted genetic ancestry provided by the *All of Us* DRC.

For predicting genetic ancestry, the All of Us Data Resource Center extracted variants from the exon regions of all autosomal, basic, protein-coding transcripts in GENCODE v42, for a training dataset consisting of samples from the Human Genome Diversity Project and 1000 Genomes. These were then used to build principle components (PCs) using Hail. These PCs were then used as the features for a random forest classifier. Samples were assigned an ancestry group if the classification probability exceeded 90%. The ancestry super-populations correspond to the ancestry definitions used within gnomAD[25], the Human Genome Diversity Project[31,32], and 1000 Genomes[6]. These include African/African American (afr), American Admixed/Latino (amr), East Asian (eas), European (eur), Middle Eastern (mid), South Asian (sas), and Other (oth; not unambiguously clustering with super-population in the principal component analysis)[31]. Concordance between self-reported race/ethnicity and these ancestry predictions is 0.898. For full information see https://support.researchallofus.org/hc/en-us/article_attachments/14969477805460/All_Of_Us_Q2_2022_Release_Genomic_Quality_Report__1_.pdf.

To detect predicted loss of function (pLoF) variants, we additionally annotated aggregate variant data using Variant Effect Predictor with the LOFTEE plugin[25]. pLoF variants were filtered, retaining only those with 'high confidence'. The following variant effects were treated as loss of function: "frameshift_variant", "stop_gained", "stop_lost", "splice_acceptor_variant, "splice_donor_variant", and "start_lost" when they were seen in the "vep.most_severe_consequence" field. Other Variant Effect Predictor "HIGH" impact effects such as "transcript_ablation"[33] were not present in the dataset.

**Statistics and reproducibility**. The comparison of pathogenic variant counts between predicted genetic ancestry groups, in the *All of Us* dataset, was done with a Chi-Square test for independence. To meet the Chi-square requirement that all entries have at least a count of 5, the less-represented ancestries (East Asian, Middle Eastern, South Asian and Other) and genes were aggregated into a single column and row in the contingency table, respectively (Supplementary Table 1). To detect genes with outlying rates of P/LP variants, we compared proportions to the European ancestry rate under the null hypothesis that the proportions are equal. We used a Z-test statistic:

$$Z = \frac{(\hat{p}_1 - \hat{p}_2)}{\sqrt{\hat{p}(1-\hat{p})(\frac{1}{n_1} + \frac{1}{n_2})}} \quad (1)$$

Where $\hat{p}$ is the overall proportion:

$$\hat{p} = \frac{Y_1 + Y_2}{n_1 + n_2} \quad (2)$$

$Y_1$ and $Y_2$ are the group (i.e., participants with primarily European ancestry vs participants with primarily African ancestry) variant counts, $n_1$ and $n_2$ are the group totals and $\hat{p}_1$ and $\hat{p}_2$ are the group proportions. To translate the Z-score to a *p* value we assumed a two-tailed normal distribution.

ClinVar comparisons used variant_summary.txt from Jan, 22, 2022, downloaded from the ClinVar downloads site[34]. We preprocessed this file by (1) filtering out GRCh37 data from the downloaded file and using GRCh38 entries only and (2) filtering the variants 2 stars and above where the ClinicalSignificance field is "Likely pathogenic" or "Likely pathogenic, risk factor" or "Pathogenic" or "Pathogenic/Likely pathogenic" or "Pathogenic/Likely pathogenic, risk factor" or "Pathogenic, risk factor" and the

ReviewStatus field is "criteria provided, multiple submitters, no conflicts", and the LastEvaluated date is on or after Jan 1, 2016. We included three and four star pathogenic variants ("reviewed by expert panel" and "practice guideline" respectively) regardless of the last evaluated time.

GnomAD data used gnomAD v2.1.1 liftover data set from the download site[35]. For ClinVar, we chose entries from 2016 or later and having two or more stars. We joined the gnomAD and ClinVar dataset by using a variant's chromosome-position-ref-alt combination as a primary key. We then calculated the pathogenic and likely pathogenic (P/LP) ratio in each gene by adding up the alternate allele count in each gene as the numerator and used the maximum of total alleles in each gene as the denominator. We calculated P/LP ratio/frequency in each gene in the dataset, building a contingency table that mirrored the format derived from that from the *All of Us* variant data.

**Reporting summary**. Further information on research design is available in the Nature Portfolio Reporting Summary linked to this article.

## Data availability

All sequencing data used in this study are available on the *All of Us* Researcher Workbench in the v6 release. Researchers can register to access this resource at: https://www.researchallofus.org/. The VIP database of curated variants is available on gitlab: https://gitlab.com/bcm-hgsc/neptune. Source data underlying Figs. 1–3 are provided in Supplementary Data 2.

## Code availability

All code used to carry out this project reside in the *All of Us* Researcher Workbench in the 'Demo - Assessment of pathogenic variants across the All of Us Research Program': https://www.researchallofus.org/.

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

## Acknowledgements

The All of Us Research Program is supported by the National Institutes of Health, Office of the Director: Regional Medical Centers: 1 OT2 OD026549; 1 OT2 OD026554; 1 OT2 OD026557; 1 OT2 OD026556; 1 OT2 OD026550; 1 OT2 OD 026552; 1 OT2 OD026553; 1 OT2 OD026548; 1 OT2 OD026551; 1 OT2 OD026555; IAA #: AOD 16037; Federally Qualified Health Centers: HHSN 263201600085U; Data and Research Center: 5 U2C OD023196; Biobank: 1 U24 OD023121; The Participant Center: U24 OD023176; Participant Technology Systems Center: 1 U24 OD023163; Communications and Engagement: 3 OT2 OD023205; 3 OT2 OD023206; and Community Partners: 1 OT2 OD025277; 3 OT2 OD025315; 1 OT2 OD025337; 1 OT2 OD025276. In addition, the All of Us Research Program would not be possible without the partnership of its participants. We thank our colleagues, Kelsey Mayo, Ashley Able, Ashley Green, Andrea Ramirez, and Sokny Lim for providing their support and input throughout the demonstration project lifecycle. We thank Dr. Jun Qian and Dr. Lina Sulieman for providing input on the project's code review. We thank Jennifer Zhang and the DRC Genomics Curation Teams for providing the data artifacts used for the project. We would also like to acknowledge the work of the late Professor Deborah A. Nickerson, who was a key member in the early phases of this project. We thank the DRC's Research Support team for their help during implementation. We also thank the *All of Us* Science Committee and *All of Us* Steering Committee for their efforts evaluating and finalizing the approved demonstration projects. The *All of Us* Research Program would not be possible without the partnership of contributions made by its participants. See below for a roster of past and present *All of Us* principle investigators. To learn more about the *All of Us* Research Program's research data repository, please visit https://www.researchallofus.org/.

## Author contributions
Conceived and designed the analysis: E.V., A.M., R.G., G.J., P.E. Collected the data: E.V., D.K., K.P., M.W., Y.C., L.L. Performed the analysis: E.V., K.P., D.K., M.W., Y.C. Wrote the paper: E.V., K.P., D.K., M.W., Y.C., S.K., B.Y., J.K., K.W., J.S., S.M., A.R., A.H., P.E., Q.W., L.L., D.T., G.J., A.M., R.G.

## Competing interests
E.V. owns shares in Codified Genomics, a provider of genetic interpretation software. All BCM-affiliated authors declare that Baylor Genetics is a BCM affiliate that derives revenue from genetic testing. All other authors declare no competing interests.

## Additional information

## the All of Us Research Program Investigators

Brian Ahmedani[11], Christine D. Cole Johnson[11], Habib Ahsan[12], Hoda Anton-Culver[13], Eric Topol[14], Katie Baca-Motes[14], Julia Moore-Vogel[14], Praduman Jain[15], Mark Begale[15], Neeta Jain[15], David Klein[15], Scott Sutherland[15], Bruce Korf[16], Beth Lewis[16], Ali G. Gharavi[17], George Hripcsak[17], Eric Boerwinkle[18], Scott Joseph Hebbring[19], Elizabeth Burnside[20], Dorothy Farrar-Edwards[20], Amy Taylor[21], Liliana Lombardi Desa[22], Steve Thibodeau[23], Mine Cicek[23], Eric Schlueter[24], Beverly Wilson Holmes[24], Martha Daviglus[25], Paul Harris[26], Consuelo Wilkins[26], Dan Roden[26], Kim Doheny[27], Evan Eichler[28], Gail Jarvik[28], Gretchen Funk[29], Anthony Philippakis[30], Heidi Rehm[30], Stacey Gabriel[30], Richard Gibbs[31], Edgar M. Gil Rico[32], David Glazer[33], Jessica Burke[34], Philip Greenland[35], Elizabeth Shenkman[36], William R. Hogan[36], Priscilla Igho-Pemu[37], Elizabeth W. Karlson[38], Jordan Smoller[38], Shawn N. Murphy[38], Margaret Elizabeth Ross[39], Rainu Kaushal[39], Eboni Winford[40], Vik Kheterpal[41], Francisco A. Moreno[42], Cheryl Thomas[43], Mitchell Lunn[44], Juno Obedin-Maliver[44], Oscar Marroquin[45], Shyam Visweswaran[45], Steven Reis[45], Patrick McGovern[46], Gregory Talavera[47], George T. O'Connor[48], Lucila Ohno-Machado[49], Fornessa Randal[50], Andreas A. Theodorou[51], Eric Reiman[51], Mercedita Roxas-Murray[52], Louisa Stark[53], Ronnie Tepp[54], Alicia Zhou[55], Scott Topper[55], Rhonda Trousdale[56], Phil Tsao[57], Scott T. Weiss[58], Jeffrey Whittle[59], Stephan Zuchner[60], Olveen Carrasquillo[60], Megan Lewis[61], Jen Uhrig[61], May Okihiro[62], Maria Argos[25], Brisa Aschebook-Kilfoy[25], Laura Bartlett[63], Roberta Carlin[64], Elizabeth Cohn[65], Vivian Colon-Lopez[66], Karl Cooper[64], Linda Cottler[67], Errol Crook[68], Elizabeth Culler[69], Charles Drum[64], Milton Eder[67], Mark Edmunds[70], Rachel Everhart[71], Adolph Falcon[32], Becky Fein[72], Zeno Frano[59], Michael Garrett[73], Sandra Halverson[74], Eileen Handberg[36], Joyce Ho[35], Laura Horne[72], Rosario Isasi[60], Jessica Isom[75], Jessica Jarmin[76], Megan Jula[77], Royan Kamyar[78], Frida Kleiman[65], Isaac Kohane[79], Babbette Lamarca[73], Brendan Lee[31], Niall Lennon[30], Dessie Levy[80], Todd Mahr[81], Emily Makahi[62], Vivienne Marshall[82], Elizabeth Mayer-Davis[83], Jacob McCauley[60], Jeffrey McKinney[84], David McPherson[18], Robert Meller[37], Jose Melo[66], David Ming-Hung Lin[85], Michael Minor[80], Evan Muse[14], Kapil Parakh[86], Cathryn Peltz-Rauchman[11], Linda Perez Laras[87], Subhara Raveendran[88], Gail Reilly[40], Jody Reilly[89], Nelida Rivera[87], Laura Rosales[31], Tracie Rosser[90], Linda Salgin[47], Sherilyn Sawyer[91], William Simonson[92], Amy Sitapati[49], Cynthia So-Armah[75], Gene Stegeman[93], Christin Suver[94], Michael Taitel[95], Kyla Taylor[40], Daniel Hernandez Tinoco[40], Scott Topper[55], Rhonda Trousdale[56], Jason Vassy[91], Jamie Walz[84], Preston Watkins[96], Blaker Wilkerson[97], Katrina Yamazaki[21], Melissa Basford[26],

Amaryllis Silva Boschetti[49], Matthew Breeden[98], Suchitra Chandrasekaran[90], Cheryl Clark[58], Kim Enard[98], Yuri Fresko[89], Richard Grucza[98], Robert Kelley[90], Kathleen Keogh[22], Monica Kraft[99], Christopher Lough[100], Ted Malmstrom[98], Paul Nemeskal[75], Matt Pagel[90], Jeffrey Scherrer[98], Sanjay Skukla[19], Debra Smith[101], Bryce Turner[102] & Miriam Vos[90]

[11]Henry Ford Health System, Detroit, MI, USA. [12]University of Chicago Medical Center, Chicago, IL, USA. [13]University of California, Irvine - Irvine, California, CA, USA. [14]Scripps Research Translational Institute - La Jolla, California, CA, USA. [15]Vibrent Health, Fairfax, VA, USA. [16]University of Alabama at Birmingham, Birmingham, AL, USA. [17]Columbia University - New York City, New York, NY, USA. [18]University of Texas Health Science Center at Houston, Houston, TX, USA. [19]Marshfield Clinic Research Institute, Marshfield, WI, USA. [20]University of Wisconsin at Madison, Madison, WI, USA. [21]Community Health Center, Inc., Middletown, CT, USA. [22]Sun River Health - Beacon, New York, NY, USA. [23]Mayo Clinic and Foundation, Rochester, MN, USA. [24]Cooperative Health, South Carolina, SC, USA. [25]University of Illinois at Chicago, Evanston, IL, USA. [26]Vanderbilt University Medical Center, Nashville, TN, USA. [27]Johns Hopkins University School of Medicine, Baltimore, MD, USA. [28]University of Washington, Seattle, Washington, WA, USA. [29]FiftyForward, Nashville, TN, USA. [30]Broad Institute - Boston Massachusetts, Boston, USA. [31]Baylor College of Medicine, Houston, TX, USA. [32]National Alliance for Hispanic Health, Washington, DC, USA. [33]Verily Life Sciences - South San Francisco, California, CA, USA. [34]MITRE Corporation, McLean, VA, USA. [35]Northwestern University, Evanston, IL, USA. [36]University of Florida, Gainesville, FL, USA. [37]Morehouse School of Medicine, Atlanta, GA, USA. [38]Partners Health Care - Boston, Massachusetts, MA, USA. [39]Cornell University, Weill Medical College - New York City, New York, NY, USA. [40]Cherokee Health Systems, Knoxville, TN, USA. [41]CareEvolution, Inc, Ann Arbor, MI, USA. [42]University of Arizona, Tucson, Tucson, AZ, USA. [43]Delta Research and Educational Foundation, Washington, DC, USA. [44]Stanford University - Stanford, California, CA, USA. [45]University of Pittsburgh, Pittsburgh, PA, USA. [46]Wondros - Los Angeles, California, CA, USA. [47]San Ysidro Health Center - San Ysidro, California, CA, USA. [48]Boston Medical Center - Boston, Massachusetts, MA, USA. [49]University of California, San Diego – San Diego, California, CA, USA. [50]Asian Health Coalition, Chicago, IL, USA. [51]Banner Health, Phoenix, AZ, USA. [52]Montage Marketing Group, Rockville, MD, USA. [53]University of Utah, Salt Lake City, Utah, USA. [54]HCM Strategists, Austin, TX, USA. [55]Color Genomics, Inc. – Burlingame, California, CA, USA. [56]NYC Health + Hospitals - New York City, New York, NY, USA. [57]VA AoU Coordinating Center - Palo Alto, California, CA, USA. [58]Brigham and Women's Hospital – Boston, Massachusetts, MA, USA. [59]Medical College of Wisconsin, Milwaukee, WI, USA. [60]University of Miami School of Medicine, Miami, FL, USA. [61]Research Triangle Institute - Research Triangle Park, Raleigh, NC, USA. [62]Waianae Coast CHC, Waianae, HI, USA. [63]National Library of Medicine (NLM), Bethesda, MD, USA. [64]American Association of Health and Disability, Sedona, AZ, USA. [65]Hunter College - New York City, New York, NY, USA. [66]University of Puerto Rico Comprehensive Cancer Center, San Juan, WA, USA. [67]CTSA Community Engagement Programs, Gainesville, FL, USA. [68]University of South Alabama, Mobile, AL, USA. [69]TPC: Blood Assurance - Signal Mountain, Tennessee, TN, USA. [70]San Diego Blood Bank – San Diego, California, CA, USA. [71]TPC: Denver Health - Denver, Colorado, CO, USA. [72]TPC: Active Minds -, Washington, DC, USA. [73]University of Mississippi Medical Center, Jackson, MS, USA. [74]TPC: DLH Corp, Atlanta, GA, USA. [75]Mass General Hospital - Boston, Massachusetts, MA, USA. [76]Tactis -, Washington, DC, USA. [77]TPC: Mary's Center -, Washington, DC, USA. [78]TPC: Owaves -, Washington, DC, USA. [79]Harvard Medical School - Boston, Massachusetts, MA, USA. [80]National Baptist Convention, Nashville, TN, USA. [81]Gundersen Health System, La Crosse, WI, USA. [82]South Texas Blood and Tissue Center, San Antonio, TX, USA. [83]University of North Carolina at Chapel Hill - Chapel Hill, Chapel Hill, NC, USA. [84]Sensis - Glendale, California, CA, USA. [85]TPC: Bloodworks Northwest – Seattle, Washington, WA, USA. [86]TPC: Fitbit - San Francisco, California, CA, USA. [87]COSSMA, Aibonito, PR, Puerto Rico. [88]Patients Like Me, Houston, TX, USA. [89]Quest Diagnostics Incorporated – Secaucus, Secaucus, NJ, USA. [90]Emory University, Atlanta, GA, USA. [91]VA AoU Coordinating Center - Boston, Massachusetts, MA, USA. [92]Cascade Regional Blood Services - Tacoma, Washington, WA, USA. [93]ExamOne, Mission, KS, USA. [94]Sage Bionetworks – Seattle, Washington, WA, USA. [95]Walgreen Co., Deerfield, IL, USA. [96]WebMD Health Corp – New York City, New York, NY, USA. [97]Blue Cross Blue Shield, Chicago, IL, USA. [98]Saint Louis University, Saint Louis, MO, USA. [99]Mount Sinai Health System - New York City, New York, NY, USA. [100]LifeSouth, Gainesville, FL, USA. [101]SunCoast Blood Center, Bradenton, FL, USA. [102]University Southern California - Los Angeles, California, CA, USA.

