## [Peer Review File · Communications Biology]

Reviewers' comments:

Reviewer #1 (Remarks to the Author):

General comments

Understudied/ underrepresented populations and particularly African genomes harbor millions of uncaptured variants accumulated over 300,000 years of modern humans' evolutionary history, with successive waves of admixture, migration, and natural selection an exceptional genomic complexity. Moreover, population in South America with native Indian ancestry might have seen their genome going through an evolutionary bottleneck, although recently admixture with European have reshaped these genomes. It is now clear that the detailed study of these underrepresented genomic variations globally is a Scientific imperative, that will inform both the genetic evolutionary history of humans, and our understanding of diseases.

Therefore, the research reported from Venner et al centered on evolving findings from the All of Us cohort will add to much needed publicly available data from diverse genomes.

The paper provides unique comparative data on pathogenic variant proportion across populations highlighting the possible disparity that could be associated to bias in ascertainment. This is particularly supported, by the differential frequency of putative pathogenic novel variants causing amino acid changes or stop codon gain or loss changes, across populations which might be suggestive, in some cases, of variant misclassifications in ClinVar and other databases.

The article is important to the global scientific community, but could be improve in a few areas.

Specific comments

1. The finding of novel ancestral divergence for gene such as PKP2 is important. I will suggest the authors provide data other well-known variants with ancestral divergence (beside HFE and PALB2) e.g. African/middle eastern ancestry: 1) sickle cell disease mutation (rs334); 2- G6PD A- 202A and 376G alleles, APOL1 G1 and G2 alleles...

2. Related to the above, a curiosity might be to investigate if current in silico tools will labelled well known rs334 as "pathogenic", owing to it huge differential frequencies across populations linked to malaria endemicity

3. Authors should discuss possible solutions and algorithm designs of variant interpretations, beside the inclusion of "diverse populations", to alleviate these potential limitations in current variants interpretations. How often in vitro functional analysis will be necessary? What number of ethnic specific apparently healthy "control" population are need to be investigated for pathogenicity of novel variants? Etc.

4. In line with the ancestral divergence of some medically actionable variants, authors should comment the relevance of American College of Medical Genetics and Genomics (ACMG) Secondary Findings gene panel (ACMG 2.0), to All populations

5. Abstract should be aligned to the text, including proportion for pathogenic variants among African and Latino, who are second and third largest population groups, respectively

6. Authors should provide limitation of their current study.

Reviewer #2 (Remarks to the Author):

Please refer to the attached PDF.

Reviewer #3 (Remarks to the Author):

In this paper Venner et al present a comparison of the frequencies of pathogenic mutations in selected genes across different ancestry groups in the All of US cohort.

The work is important with significant potential for translation. All of US also represents an exciting novel data resource that is likely to significantly impact on medical genomics. There are several potential sources for the differences and it is extremely important to clarify what causes them.

The authors use two selected list of genes and different ways of annotating likely pathogenic mutations. They then compare rates of these mutations across ancestry groups. Differences are interpreted as indicating potential ancestry differences in mutation rates.

The initial analysis of curated (?) variants finds higher rates of pathogenic mutations in the European ancestry group. More information about how these variants were annotated would be vital here. This is not even present in the methods. What is the VIP data base and how were variants annotated as "P/LP" there?

In a second analyses they include all predicted loss of function variants that are rare in a reference cohort. The authors find that differences between ancestry groups decrease. However, I cannot find the results from the very important second analyses comprehensively shown. The main figure only has the results from the first analysis. Moreover, the differences in variant annotation as pathogenic or not should be clarified. The finding that variant annotation based on predominantly European ancestry data may lead to under-diagnosis of people from other ancestry groups would be quite significant.

In the second analysis, variants were filtered based on Gnomad allele frequencies. The largest ancestry group in Gnomad are European ancestry. This implies better filtering out of non-pathogenic variants for variants sufficiently present in European ancestry participants. So it could artificially increase the number of potential pathogenic variants in other ancestry groups. This could bias the comparison between the two approaches.

It would be good to pay careful attention to labelling when it comes to ancestry. For example, instead of "European subgroup" there is a European ancestry subgroup. "Other" is a difficult label.

Ancestry assignment for the AMR group and labelling them as Latin is tricky. This is not a genetically defined superpopulation. There is much admixture. So identifying participants of genetically as being Latin American is problematic.

Not in the power of the authors to change the ACMG gene list but I was a bit surprised not to find HBB, the cause of the most common group of genetic disorders. Related to this, a point for discussion is whether the selection of genes for these lists may also be biased to genes with more frequent mutation rates in participants of European ancestry.

I am not comfortable with the penetrance calculation. Properly done would be a lot more complex. The current analysis completely ignores participant age, censoring, preventative measures such as mastectomy etc. I would strongly advise against reporting this at all. Penetrance have significant clinical impact so should not be reported without careful consideration.

We would like to thank the reviewers for their feedback. The paper has markedly improved as a result. Our responses to the reviewer's feedback are below, with reviewer / editor comments in **bold**, our responses in plan text and our revised text in blockquotes with new text in blue.

As part of addressing the reviewer's comments we have fully redone all analyses, figures and tables in this paper. The All of Us data analyzed is now the v6 release, which contains a similar number of participants but uses a genetic ancestry calculation that considers all exonic regions as opposed to only chromosome 20 & 21. As a result, the data have changed slightly but the conclusions remain the same.

Reviewers' comments:

Reviewer #1 (Remarks to the Author):

General comments

Understudied/ underrepresented populations and particularly African genomes harbor millions of uncaptured variants accumulated over 300,000 years of modern humans' evolutionary history, with successive waves of admixture, migration, and natural selection an exceptional genomic complexity. Moreover, population in South America with native Indian ancestry might have seen their genome going through an evolutionary bottleneck, although recently admixture with European have reshaped these genomes. It is now clear that the detailed study of these underrepresented genomic variations globally is a Scientific imperative, that will inform both the genetic evolutionary history of humans, and our understanding of diseases.

Therefore, the research reported from Venner et al centered on evolving findings from the All of Us cohort will add to much needed publicly available data from diverse genomes.

The paper provides unique comparative data on pathogenic variant proportion across populations highlighting the possible disparity that could be associated to bias in ascertainment. This is particularly supported, by the differential frequency of putative pathogenic novel variants causing amino acid changes or stop codon gain or loss changes, across populations which might be suggestive, in some cases, of variant misclassifications in ClinVar and other databases.

The article is important to the global scientific community, but could be improve in a few areas.

Specific comments

1. The finding of novel ancestral divergence for gene such as PKP2 is important. I will suggest the authors provide data other well-known variants with ancestral divergence (beside HFE and

PALB2) e.g. African/middle eastern ancestry: 1) sickle cell disease mutation (rs334); 2- G6PD A- 202A and 376G alleles, APOL1 G1 and G2 alleles...

A separate manuscript is in preparation, which will describe the pharmacogenomic results, including analysis of G6PD alleles, broken down by ancestry. We have included an additional analysis of APOL1 and HBB. The results show the expected enrichments in the African ancestry.

In Results, we have added the following new paragraph:

In order to evaluate whether known variants with ancestral divergence are replicated in the All of Us cohort, we examined the frequency of the rs334 mutation in SBB, known to be associated with sickle cell disease, and the APOL1 G1 and G2 alleles. The results confirm the expected ancestral divergences, with non-reference alleles appearing 17,969 times within the 22,897 participants of African ancestry, and 68 times among the 49,668 participants of European ancestry.

As well as the following new table:

	African	Latino / Admixed American	East Asian	European	Middle Eastern	Other	South Asian
APOL1 (G1 & G2 alleles)	17,969 / 22,897 (78.48%)	1,041 / 15,893 (6.55%)	1 / 2,113 (0.05%)	60 / 49,668 (0.12%)	1 / 193 (0.52%)	1,166 / 6,886 (16.93%)	2 / 940 (0.21%)
HBB (rs334)	2059 / 22,897 (8.99%)	229 / 15,893 (1.44%)	0 / 2,113 (0.00%)	6 / 49,668 (0.01%)	0 / 193 (0.00%)	202 / 6,886 (2.93%)	2 / 940 (0.21%)

Table 1: Non-reference allele counts known ancestrally-divergent genes. To confirm the presence of known alleles with ancestral divergence, we examined the APOL1 G1 and G2 alleles, as well as the rs334 SNP in the HBB gene. The counts in the table above include all instances of the rs73885319, rs60910145 SNPS (APOL1 G1 allele) as well as the G2 deletion (rs71785313) and the rs334 allele in the HBB gene, associated with Sickle-cell disease. Participants may carry more than one allele. These variants are present at a higher rate in participants with African ancestry, as expected.

2. Related to the above, a curiosity might be to investigate if current in silico tools will labelled well known rs334 as “pathogenic”, owing to it huge differential frequencies across populations linked to malaria endemicity

We have not reported the HBB gene in our clinical lab, so the rs334 SNP does not appear in our VIP database. Our automated variant classification tools do not make a specific prediction for the pathogenicity of this variant. We do however link to ClinVar and note the “Conflicting interpretations of pathogenicity”. If a reviewer was unfamiliar with the variant, that would give them a place to begin their research.

3. Authors should discuss possible solutions and algorithm designs of variant interpretations, beside the inclusion of “diverse populations”, to alleviate these potential limitations in current variants interpretations. How often in vitro functional analysis will be necessary? What number of ethnic specific apparently healthy “control” population are need to be investigated for pathogenicity of novel variants? Etc.

We have added the following paragraph in the discussion:

It is paramount that the field address ascertainment biases in knowledge of pathogenic variants. Case-control data are a powerful tool towards achieving this, and the field would benefit greatly from creating more diverse cohorts of patients with accompanying variant interpretation, and from sharing those interpretations widely. As new VUS are identified, functional studies are also beneficial in their interpretation. The advent of high-throughput functional screening techniques is accelerating[<https://doi.org/10.1038/s43586-021-00093-4>] this data collection and could be targeted at variants from underrepresented populations.

4. In line with the ancestral divergence of some medically actionable variants, authors should comment the relevance of American College of Medical Genetics and Genomics (ACMG) Secondary Findings gene panel (ACMG 2.0), to All populations

We have additionally added this paragraph to the discussion:

An alternative, although unlikely, explanation for the elevated rate of pathogenic variants seen in this study in the European population, is that the selected genes carry a burden of pathogenic variants that is in fact specific to individuals of European ancestry. The ACMG took an evidence-based approach [<https://www.nature.com/articles/s41436-021-01172-3>] to selecting genes, prioritizing genes which, at the time, had sufficient evidence showing that patient morbidity and mortality could be reduced through genetic testing while limiting the burden on patients and clinical laboratories. Given known disparities in genetics knowledge, this process may have preferentially identified genes whose impacts on individuals with European ancestry is well studied. Remediating this will require future case-control studies that include diverse populations, which expert panels can then review as they decide on future secondary finding lists.

5. Abstract should be aligned to the text, including proportion for pathogenic variants among African and Latino, who are second and third largest population groups, respectively

We have made the following update to the abstract:

The European subgroup showed the highest overall rate of pathogenic variation, with 2.13% of participants having a known pathogenic variant. Other ancestry groups had

lower rates of pathogenic variation, including 1.52% for the African ancestry group and 1.17% in the Latino/Admixed American group.

6. Authors should provide limitation of their current study.

We have added the following paragraph in the discussion section:

The current study faces a number of limitations. First, our variant knowledge relies on variant interpretations that were primarily carried out for other genomic reporting projects and during the lead-up to the return of health-related results for the program. As we complete the health-related return of results and the accompanying variant reviews, we will also curate new variants, which is likely to benefit the underserved populations. We are also limited by the cohort creation process adhered to by the all of us research program. This process recruits participants through partnerships with universities, research centers, and community health centers, direct volunteers, and through community engagement. This process will undoubtedly shape the cohort; for example, we know that participants with a family history of disease are more likely to sign up for genetics studies [<https://link.springer.com/article/10.1007/s12687-018-0375-3>]. At this time, it is not clear what effect this selection process would have on the rates of rare, pathogenic variants. As evidenced by our examination of selected known pathogenic variants and of the high-level comparisons with GnomAD, there does not appear to be a large selection bias. Further study is needed in order to understand whether there are localized or more subtle selection effects at work in this cohort.

Reviewer #2 (Remarks to the Author):

Reviewer's comment on the manuscript (MS # COMMSBIO-22-2650-T) "The Frequency of Pathogenic Variation in the All of Us Cohort Reveals Ancestry-driven Disparities"

In this manuscript, the authors examined the frequency of pathogenic variants in 73 genes conferring significant health risks in the All of Us cohort. Compared to the other population projects such as the UK Biobank and the Million Veterans project, the All of Us Research Program which provides whole-genome sequencing, surveys, electronic health records, and physical measurements data has a higher proportion of nonEuropean descendent participants and higher ancestral diversity of the participants. The results show that some pathogenic variants have different frequency spectra across the study ancestral groups ("afr", "amr", "eas", "eur", "mid", "sas", and "oth"). The authors conclude that the differences in the frequency of pathogenic variants in the different ancestral groups indicate biases of ascertainment in this All of Us cohort.

Ascertainment bias is a potential confounder that may cause the results in the analysis to be misleading. Please provide an evaluation and analysis for drawing this conclusion in this All of Us cohort, and then evaluate the intrinsic difference in the frequency of pathogenic variants in the different ancestral groups after adjusting for the

ascertainment bias. Please demonstrate control of ascertainment bias in this study to prevent misleading findings, particularly for cross-ancestry comparisons.

We agree that ascertainment bias is a potential cofounder, and previous studies have noted that, given the choice, participants with a known family history of genetic disease will preferentially sign up for genetics studies [10.1007/s12687-018-0375-3]. However, we see no evidence that these types of biases are impacting the rates of genetic variation that explain rare Mendelian disease. In addition to the GnomAD comparison, where we found that the All of Us rates were generally concordant with those from GnomAD and in some specific examples with discrepancies (e.g. African *BRCA2*, Latino / Admixed American *APOB*), the All of Us numbers were closer to the unselected portion of GnomAD (non-cancer and non-TopMed respectively).

To further check whether we could detect evidence of ascertainment bias, we examined a set of specific variants, ranging from very rare to relatively common, and compared the frequencies with those from GnomAD. We find remarkably close allele frequencies, with only the difference in the common HFE variant rs1800562 rising to significance.

Because of this, within the All of Us cohort, we do not believe that there is a large effect from ascertaining participants with genetic conditions. We do recognize that there is likely some selection effect present in the cohort, and that it may have a small effect size that is hard to measure due to dependence on specifics of the participants ancestry, disease and other factors.

We have added the following new table and Results paragraph.

Variant	GnomAD AF	All of Us AF	p value (Two-tailed z test)	Population
BRCA2 c.5946delT (p.Ser1982Argfs*22)	0.0364%	0.0342%	0.815	Eur
BRCA2 c.2808_2811delAAAG (p.Lys938Ilefs*7)	0.0014%	0.0020%	0.703	Eur
BRCA2 c.8537_8538delAG (p.Ser2846Glufs*2)	0.0021%	0.0010%	0.525	Eur
LDLR c.682G>T (p.Glu228X)	0.0024%	0.0050%	0.375	Eur
APOL1 G1 p.S342G - rs73885319 (GRCh38:chr22:36265860:A>G)	22.27%	22.36%	0.753	Afr
APOL1 G1 p.I384M - rs60910145 (GRCh38:chr22:36265988:T>C/G)	22.43%	21.94%	0.081	Afr
HBB rs334	4.34%	4.50%	0.269	Afr
HFE rs1800562	6.05%	6.41%	0.001	Eur

Table 3: Comparison of allele frequencies for specific variants with GnomAD. To evaluate whether self-selection by participants with known genetic diseases impacts our study, we examined the allele frequencies of eight disease-associated variants (four rare, four common) in comparison with GnomAD. Except for a 6% difference in the common HFE rs1800562 alleles, the frequencies in our study closely match those in GnomAD.

In Results:

As an additional approach to understand whether *All of Us* participants with known genetic diseases are self-selecting for participation in the program, which could impact the counts of pathogenic variants impacting Mendelian disease. We selected four rare and four more common variants associated with disease, and examined their allele frequencies relative to that in GnomAD (Table 3). To best match the *All of Us* ancestry groups, the GnomAD Eur group was taken as a combination of Finish, non-Finish and Ashkenazi populations. For these variants, we observe a close match between their GnomAD and *All of Us* allele frequencies. Only the common HFE rs1800562 alleles, in which a 6% difference was found, was significantly different (Bonferroni corrected critical value of 0.00625). Based on this analysis, we do not find an appreciable amount of ascertainment bias in this population.

We have also mentioned this concern in the limitations section of the Discussion:

Although we have not been able to detect selection bias in the cohort in this analysis, the cohort likely does contain both enrichments and depletions of pathogenic variants in specific disease genes due to self-selection of the participants. These effects may be local to specific disease areas or specific subpopulations within the cohort. Researchers making use of this resource should be aware of this potential bias.

As an additional note, in this manuscript, we have used the term ‘ascertainment bias’ to refer to the ascertainment of variant knowledge, which to date is biased towards variants found in the European population. We have clarified this point in the Manuscript:

Differences in the frequency of pathogenic variants observed between ancestral groups generally indicate biases of ascertainment of knowledge about those variants

Future work will show whether variant interpretation of the *All of Us* diverse cohort will have an impact on this ascertainment bias of pathogenic variants

The definition of ancestry groups is crucial in this study. Genetic ancestry was inferred by using random forest based on the first 16 principal components. The principal components were calculated based on only 56,671 variants from chromosomes 20 and 21. Please explain why didn't include high-quality variants from all autosomes in the current analysis. Any differences in the results of the analysis based only on chromosomes 20 and 21 and the analysis based on all autosomes? Will the differences influence the findings in this study? Were the variants with linkage disequilibrium removed before the construction of the principal components? If not, the obtained principal components may be biased. Random forest was applied to classify the participants in the All of Us cohort into different ancestry super-populations based on the first 16 principal components. True ancestral groups should be provided when establishing a training model in a supervised random forest classification analysis. Did the true answer come from self-reported ancestry or other information? Please elaborate

more on the analysis procedure; details can be provided in a supplemental file. If self-reported ancestry was employed as the true answer in the training model, please evaluate its feasibility.

We agree and have redone the ancestry calculation and all of the subsequent analyses, tables and figures, using full genome data as a basis for the ancestry calculation instead of only chromosomes 20 and 21 (which were chosen initially for computational expediency). A substantial proportion (4%) of participants changed groups, mostly by moving from the “Other” ancestry group into a continental ancestry. These changes are exhibited in the diagram below. However, these changes resulted only in very minor changes to the counts for the number of pathogenic variants in specific ancestry groups.

After some discussion we have opted to leave information about the previous ancestry definitions out of the manuscript to avoid confusion. We have updated the methods accordingly:

In Methods:

For predicting genetic ancestry, the All of Us Data Resource Center extracted variants from the exon regions of all autosomal, basic, protein-coding transcripts in GENCODE v42, for a training dataset consisting of samples from the Human Genome Diversity Project and 1000 Genomes. These were then used to build principle components (PCs) using Hail. These PCs were then used as the features for a random forest classifier. Samples were assigned an ancestry group if the classification probability exceeded 90%. The ancestry super-populations correspond to the ancestry definitions used within gnomAD²⁶, the Human Genome Diversity Project^{30,31}, and 1000 Genomes⁶. These include African/African American (afr), American Admixed/Latino (amr), East Asian (eas), European (eur), Middle Eastern (mid), South Asian (sas), and Other (oth; not unambiguously clustering with super-population in the principal component analysis

(PCA)³⁰. Concordance between self-reported race/ethnicity and these ancestry predictions is 0.898. For full information see https://support.researchallofus.org/hc/en-us/article_attachments/14969477805460/All_of_Us_Q2_2022_Release_Genomic_Quality_Report_1.pdf

Multiple-testing corrections should be performed in this study. For example, in order to understand which genes have divergent rates of pathogenic variants between ancestry groups, the authors applied a two-proportion test (z test) to many variants and only showed the results of a few significant genes in Table 1. Because many tests were performed, multiple-testing corrections are needed to reduce an inflated false positive. Please describe the multiple-testing correction procedure in the statistical analyses section.

When we redid the analysis to address the issues with the ancestry calculation pointed out above, we also updated them to add a multiple testing correction. All previously significant results remained significant with a Bonferoni-corrected critical value of 0.0018 with the exception of the MYL3 finding in the Latino/Admixed American group.

We have updated the relevant paragraph in the results as follows:

In order to understand which genes have divergent rates of pathogenic variants between ancestry groups, we normalized pathogenic rates of all genes against the European ancestry group's pathogenic rate and checked for outliers in population proportions using a Bonferoni-corrected z-test. Several genes show significant differences from the European ancestry group's rates (Table 2).

And the Results table:

Gene	Ancestry group	Ancestry group path. variants	European group path. variants	p value
APOB	African	4 / 22,897 (0.02%)	57 / 49,668 (0.11%)	0.0001
PKP2	African	33 / 22,897 (0.14%)	20 / 49,668 (0.04%)	0.00001
PALB2	African	33 / 22,897 (0.14%)	30 / 49,668 (0.06%)	0.0014

The authors stress that the All of Us cohort has a strength of the high diversity of the participants. However, Fig. S1 shows that the majority of the participants are of European, African, and Latino ancestry in the current release. Sample sizes in Asian populations are small. For example, this may have caused the long error bar in the Middle Eastern group (Fig. 1a). Some results for the groups with a small sample size may be not stable. The complete All of Us cohort of 1 million participants has the strength of

high diversity, but the current beta release (alpha3) of 98,000 participants does not. This issue should be reminded.

This is correct. We have added the following text to the limitations section in the discussion:

Although the full cohort for the All of Us project is anticipated to be the most diverse genetic resource available, the current release still features a large proportion of participants of European ancestry. Future releases will enable further exploration beyond what is possible in the current release.

The 1000 Genomes project (PMID: 26432245) recruited about 2,500 participants from diverse ancestral groups and also provided whole-genome sequencing data. It will be of interest to know the similarities and differences between the 1000 Genomes project and the beta release (alpha3) of 98,000 participants in the All of Us cohort.

Statistically significant comparisons to the 1000 Genomes project are difficult, especially for ancestry groups with less representation, due to the small sample size of the project. The comparison is further complicated by a lack of information in the 1000 genomes paper about the genes used or the criteria used for selecting variants, other than to state that the variants are implicated in rare disease.

The 1000 genomes publication limits discussion of these results to the following sentence: "...and 24–30 variants per genome implicated in rare disease through ClinVar; with European ancestry genomes at the high-end of these counts." This observation is largely concordant with our data from the All of Us project. We note also that the rate given in Table 1 for the AMR group is a large outlier, and this outlier is not depicted in Extended Data Figure 4, nor is it commented on in the manuscript or the supplementary information, and thus may be a typo in the table. For this reason, we have chosen not to include specific discussion of the 1000 Genomes AMR value in the manuscript here.

AFR, AMR, EAS, EUR and SAS

Figure 1: rates of pathogenic variants in the All of Us cohort compared with 1000 genomes. In both cohorts the European ancestry has a relatively high count and there is considerable variability between the groups.

We have added the following text to the manuscript:

As an additional comparison for these results, we evaluated data from the 1000 genomes project[PMID: 26432245], which reported the number of rare disease ClinVar variants across their diverse cohort. Although the gene and variant selection are not identical, the two projects are concordant in observing a relatively higher rate of pathogenic variants in the European ancestry subgroup as well as variation between the subgroups.

How to interpret the result in Fig. 1b that pathogenic variants were most frequently observed in genes related to Breast/Ovarian Cancer, Hypercholesterolemia, or Hemochromatosis? Please provide an explanation of the results. Is possible that this result was influenced by the mentioned ascertainment biases? The current presentation in the manuscript is only descriptive.

It is a challenge to find an exact baseline for expectations of variant frequency, an additional comparison is with the eMERGE III project. In this project, 11 clinical sites submitted samples to two clinical laboratories for clinical reporting. The network used a custom gene panel of 109 genes, which contained the ACMG 59 secondary findings genes, plus additional selected by the sites. Critically for this comparison, a large set of samples were chosen for the program. These findings fit with expectations based on pathogenic variant prevalence in the AMP/ACMG secondary findings genes. We compiled this comparison into a new supplementary table. Overall, we observed a higher rate of pathogenic findings in the eMERGE cohort. This is likely due to different genes in the gene panel, and the fact that test design (i.e the specific set of genes that were returned) varied by clinical site. In addition, the eMERGE set was fully reviewed whereas in the All of Us data we are reporting known pathogenic variants. It is also notable that

the eMERGE cohort is less diverse than All of Us. The rate of findings has a correlation coefficient of .78 and there is a very close match for hemochromatosis, which had identical reporting criteria of the homozygous pathogenic allele C282Y. The large difference in clotting disorders is likely explained by decision to report F5 leiden variant in eMERGE III. Overall, although it is challenging to prove a negative, these results do not indicate a selection effect in the All of Us cohort, especially taken with the GnomAD comparisons conducted previously.

We have added the following new table to the supplementary information:

	eMERGE III	All of Us
Cancer susceptibility	1.38%	0.69%
Cardiac diseases	0.87%	0.29%
Cholesterol and lipid disorders	0.50%	0.38%
hemochromatosis	0.30%	0.26%
Connective tissue / clotting disorders	0.10%	0.01%
neuromuscular diseases	0.08%	0.14%
endocrine / Immunological / Metabolic diseases	0.04%	0.01%

Supplementary Table 5: comparison of rates of pathogenic findings between eMERGE III and the All of Us research program.

We have added the following paragraph to the Results section describing these observations:

In further analysis, we examined the rate of pathogenic variants as seen in the eMERGE III project. This program involved eleven clinical sites providing samples to two clinical laboratories for analysis and reporting. The network utilized a custom gene panel of 109 genes, incorporating the ACMG 59 secondary findings list along with clinical site-specific genes. Notably for this comparison, a significant set of samples from this program were not chosen based on specific diseases. The observations in the All of Us cohort align with expectations drawn from pathogenic variant frequency in the AMP/ACMG secondary findings genes (See Supplementary Table 5, correlation coefficient for the rate of findings is 0.78). On the whole, the eMERGE III cohort exhibited a higher incidence of pathogenic findings. This can possibly be attributed to the different genes present in the gene panel and the fact that test design, which is to say the particular set of genes that were returned, varied among clinical sites. Moreover, the eMERGE data underwent complete review, unlike the All of Us data, where we are citing known pathogenic variants. There is a remarkably close match for hemochromatosis, which had the same reporting criteria for the homozygous pathogenic allele C282Y. The substantial

discrepancy in clotting disorders is likely due to the decision to report the F5 leiden variant in eMERGE III.

Please provide a brief caption and description to improve the readability of the supplemental tables and figures.

We have added captions to the supplemental tables.

Reviewer #3 (Remarks to the Author):

In this paper Venner et al present a comparison of the frequencies of pathogenic mutations in selected genes across different ancestry groups in the All of US cohort.

The work is important with significant potential for translation. All of US also represents an exciting novel data resource that is likely to significantly impact on medical genomics. There are several potential sources for the differences and it is extremely important to clarify what causes them.

The authors use two selected list of genes and different ways of annotating likely pathogenic mutations. They then compare rates of these mutations across ancestry groups. Differences are interpreted as indicating potential ancestry differences in mutation rates.

The initial analysis of curated (?) variants finds higher rates of pathogenic mutations in the European ancestry group. More information about how these variants were annotated would be vital here. This is not even present in the methods. What is the VIP data base and how were variants annotated as “P/LP” there?

We clarified and have added to the description of the VIP database to the methods section:

We used the 'VIP' database to annotate known pathogenic or likely pathogenic (P/LP) variants. The VIP database is a collection of variants compiled during clinical reporting activities carried out at the Human Genome Sequencing Center-Clinical Laboratory (HGSC-CL). These variants were manually assessed by a team of variant curation experts led by board-certified clinical geneticists for previous HGSC-CL projects, such as the NIH's eMERGE III program and HeartCare, a local cardiovascular risk assessment project. All the variants were interpreted based on the guidelines provided by the Association for Molecular Pathology/American College of Medical Genetics and Genomics (AMP/ACMG), as well as the most recent ClinGen recommendations. The VIP database includes 59,405 genomic variants. These variants are classified into different categories such as pathogenic, likely pathogenic, variants of uncertain significance (VUS), benign, likely benign, and risk alleles. They are distributed across 8,653 genes (See the supplemental file vip_gene_count.xlsx for details). One of the key

advantages of the VIP database is the consistency in variant curation, as all the variants have been curated by the Clinical Variant Interpretation team at HGSC-CL using a uniform set of criteria. This contrasts with resources like ClinVar, which may include contradictory assessments of variant pathogenicity.

In a second analyses they include all predicted loss of function variants that are rare in a reference cohort. The authors find that differences between ancestry groups decrease. However, I cannot find the results from the very important second analyses comprehensively shown. The main figure only has the results from the first analysis. Moreover, the differences in variant annotation as pathogenic or not should be clarified. The finding that variant annotation based on predominantly European ancestry data may lead to under-diagnosis of people from other ancestry groups would be quite significant.

The results in Figure 1a and b do include this data but the description was not clear. We have revised the text in Results and the figure caption to clarify this data.

To gain a more comprehensive understanding of pathogenic and likely pathogenic variants in participants, we incorporated rare (GnomAD popmax allele frequency below 0.001), predicted loss-of-function (pLoF) variants into the prior analysis of known pathogenic variants. We focused on 38 specific genes where loss-of-function is a recognized cause of disease, and we also looked for any overlap with known pathogenic variation (refer to Figure 1a, orange and gray bars). Our findings revealed 1,114 variants, comprising 562 frameshifts, 112 splice acceptor variants, 100 splice donor variants, and 340 stop gain variants (see Supplemental table 3 for details). Many of these pLoF variants overlap the pathogenic/likely pathogenic variants in the VIP database (Figures 1a, 1b, orange bars). For example, in the group with European ancestry, the overlap was 0.41%. Including pLoF variants in our analysis increases the overall rates of positive variants, ranging from 2.29% in the European ancestry group to 1.18% in the Middle Eastern and East Asian ancestry groups, with smaller ancestry groups having a wider confidence interval.

When looking purely at rates of rare pLoF variants, the European ancestry group had a lower rate of findings than both the South Asian ancestry group (1.03%) and the Other ancestry group (0.65%), though this may be due to using GnomAD, which contains samples from predominantly European ancestry, as a filter. The variation of pLoF variants between different ancestry groups was less than that of the pathogenic/likely pathogenic variants (pLoF standard deviation of variant rates was 0.003, versus a standard deviation of 0.004 for P/LP variant rates). Given that interpreting novel LoF variants can be simplified and automated, these findings suggest that the detection of pathogenic variants in studies with a large number of participants with European ancestry may be contributing to some of the differences observed between groups.

In the second analysis, variants were filtered based on Gnomad allele frequencies. The largest ancestry group in Gnomad are European ancestry. This implies better filtering out

of non-pathogenic variants for variants sufficiently present in European ancestry participants. So it could artificially increase the number of potential pathogenic variants in other ancestry groups. This could bias the comparison between the two approaches.

This is an excellent point and likely has some effect on the detected pLoF variants, although we note even with this potential bias, the European ancestry subgroup still has more pLoF variants than some of the other groups. We have added the following description of to the limitations paragraph describing this potential issue:

A further limitation is in the high-throughput nature of the pLoF analysis. Mirroring AMP/ACMG PVS1 guidelines, we filtered pLoF variants by their allele frequency, keeping only those that are rare in the GnomAD database. However, this potentially introduces a bias, as GnomAD is overrepresented with European ancestry individuals. There may remain pLoF variants that would be removed from other ancestry groups with more data. Though it is currently a limitation, as diverse datasets become available, these high-throughput estimates will improve.

It would be good to pay careful attention to labelling when it comes to ancestry. For example, instead of “European subgroup” there is a European ancestry subgroup. “Other” is a difficult label. Ancestry assignment for the AMR group and labelling them as Latin is tricky. This is not a genetically defined superpopulation. There is much admixture. So identifying participants of genetically as being Latin American is problematic.

We agree and have reviewed and updated the ancestry labeling in several locations throughout the manuscript. For example:

Abstract:

“The European ancestry subgroup showed the highest...”

Introduction

“Most sequencing studies have focused on European ancestry populations...”

“In contrast, the UK Biobank project contains 94% European ancestry individuals...”

“...15,072 (15.3%) Latino/Admixed American ancestry participants...”

Results

“...for example, in the European ancestry group...”

“...ranging from 2.29% for the European ancestry group...”

“... the European ancestry group no longer has the highest rate of findings...”

“...rates of all genes against the European ancestry group’s pathogenic rate and checked for outliers in population proportions using a z-test. Several genes show significant differences from the European ancestry group’s rates...”

“ However, when comparing gene-level pathogenic rates in ancestry groups other than the European ancestry group”

“A similar situation is seen in the Admixed American / Latino ancestry group with *LDLR*”

“...to 1.18% for the Middle Eastern and East Asian ancestry groups...”

“of findings with the South Asian ancestry group (1.03%) and Other ancestry group (0.65%)”

“...the gnomAD counts do not include the Middle Eastern and South Asian ancestry groups...”

In Methods:

“...we compared proportions to the European ancestry rate...”

Tables:

“European ancestry group path. variants”

In the context of genetic studies, "Latino/Admixed American" is often used to categorize individuals who are of Latin American descent, and it reflects the mixed (admixed) ancestral origins of these populations. This usage is common in many studies, especially those using GnomAD data, for example:

[\[https://www.sciencedirect.com/science/article/abs/pii/S0303720722001964\]](https://www.sciencedirect.com/science/article/abs/pii/S0303720722001964)

We agree, however, that such labels are problematic. Ideally, the field will move to reflecting participant ancestry in a more nuanced way. We recognize that within the "Latino/Admixed American" there's a huge amount of diversity, reflecting the different mixtures of ancestral populations in different regions of Latin America. There is ongoing discussion in the scientific community about the best ways to categorize and describe human genetic diversity in a way that is both scientifically useful and respectful of individuals' identities and experiences (<https://doi.org/10.1016/j.xgen.2022.100155>). We have added the following additional paragraph to the Discussion reflecting on these issues:

Using continental ancestry faces many issues, both in their reflection of admixed individuals and in their grouping of individuals that should not be. In future work we expect to adopt more nuanced approaches to ancestry prediction. One such approach may be to use 'local ancestry', in which ancestry information is tracked at the variant level, with each variant assigned proportionally to one of many known ancestry groups.

Not in the power of the authors to change the ACMG gene list but I was a bit surprised not to find HBB, the cause of the most common group of genetic disorders. Related to this, a point for discussion is whether the selection of genes for these lists may also be biased to genes with more frequent mutation rates in participants of European ancestry.

We agree and have added an analysis of the HBB gene (see Reviewer 1, feedback item 1). Additionally, we have added a paragraph to the discussion concerning the ACMG gene selection process and its ramifications on this work (see Reviewer 1, feedback item 4).

I am not comfortable with the penetrance calculation. Properly done would be a lot more complex. The current analysis completely ignores participant age, censoring, preventative measures such as mastectomy etc. I would strongly advise against reporting this at all. Penetrance have significant clinical impact so should not be reported without careful consideration.

We agree, and have removed the penetrance calculation from the manuscript completely.

Additionally, we have made edits throughout the manuscript for clarity.

REVIEWERS' COMMENTS:

Reviewer #1 (Remarks to the Author):

The authors have properly addressed my comments. This is much better version of their manuscript.

I will recommend accepting it for publication.

Reviewer #2 (Remarks to the Author):

The authors have adequately addressed my comments raised in a previous round of review.

Reviewer #3 (Remarks to the Author):

The authors have done an excellent job revising the manuscript. They have comprehensively addressed all my concerns. I find this work highly important. The conclusions are nuanced and justified given the results.